# Evaluation of Hardness and Residual Stress Changes of AISI 4140 Steel Due to Thermal Load during Surface Grinding

**Ewald Kohls** [1,2,*], **Carsten Heinzel** [1,2] and **Marco Eich** [1,2]

1   Leibniz Institute for Materials Engineering IWT, Badgasteiner Straße 3, 28359 Bremen, Germany; heinzel@iwt.uni-bremen.de (C.H.); eich@iwt.uni-bremen.de (M.E.)
2   MAPEX Center for Materials and Processes, University of Bremen, Bibliothekstr. 1, 28359 Bremen, Germany
*   Correspondence: kohls@iwt.uni-bremen.de; Tel.: +49-421-218-51197

**Abstract:** During surface grinding, internal material loads are generated, which take effect on the surface and subsurface zone of AISI 4140 steel. High thermal loads can result in specific material modifications, e.g., hardness reduction and tensile residual stresses, due to inappropriate combinations of system and process parameters which influence the functional performance of the ground component in a negative way. In order to avoid this damaging impact due to the thermal effect, an in-depth understanding of the thermal loads and the resulting modifications is required. This relationship is described in the concept of Process Signatures applied in this paper. Experimentally determined temperature-time histories at various depths below the surface were used to estimate the thermal loads at the surface and subsurface using a numerical approach based on the finite element method (FEM). The results show that the hardness change during surface grinding correlates with the maximum temperature rate at given maximum temperatures. In addition, correlations between the hardness change and the Hollomon–Jaffe parameter are identified, taking into account both the absolute temperature and its evolution over time. Furthermore, it was shown that the surface residual stresses correlate with the maximum local temperature gradients at the surface if no detectable tempering of the microstructure takes place.

**Keywords:** grinding; thermal load; material modification; process signature

## 1. Introduction and State of the Art

The functional performance of steel components strongly depends on their surface layer properties, such as hardness, residual stress, and microstructure. Due to manufacturing processes, the surface layer and thus the functional properties can change significantly. In finishing operations such as grinding, the surface layer changes (material modifications) become even more relevant, as these processes are the last in the process chain and therefore have a decisive importance for the material properties and the surface as well as for the subsurface integrity. The effect of grinding processes on material modifications is discussed in several scientific works. It has been shown that grinding can lead to improved functional behavior of the ground component, which can usually be attributed to the mechanical impact, e.g., in grind-strengthening [1]. In contrast to this, the thermal effect on material modifications during grinding is often connected to a deterioration of functional properties, e.g., fatigue strength and component lifetime [2]. This is caused by high thermal loads, which lead to a reduction in hardness due to annealing effects, an increase in tensile residual stresses, or even phase transformations at the workpiece surface and at the workpiece subsurface.

Predicting material modifications due to manufacturing processes with thermal impact in order to avoid negative influences on the functional performance is a key challenge for many decades. The influence of different system and process parameters on material modifications is discussed in several research studies. Uhlmann and Lin investigated the

influence of these parameters for rail grinding [3,4]. Liu showed the influence of the grinding forces on surface integrity [5]. Furthermore, it was shown that the macro-topography of a grinding wheel [6], as well as the conditions of the metalworking fluid (MWF), can lead to a significant reduction in the thermal impact [7]. In further research by Rowe and Morgan et al., Jin and Stephenson, as well as Jamshidi and Budak, a simplified approach was used to investigate the heat distributed into the grinding wheel [8], into the metalworking fluid [9], and into the workpiece [10]. High temperatures during grinding can lead to the grinding burn. The research of Malkin and Heinzel determined the thermal impact in order to identify a point at which grinding burn occurs [11,12]. In order to detect grinding burn, in-process Barkhausen noise measurements were used [12]. Baumgart used a two-color pyrometer to determine the temperatures during grinding [13]. Post-process characterization of thermally affected surfaces can be done by residual stress measurements [14]. Brinksmeier showed that the specific grinding power $P_c''$, which is a process quantity, can be correlated with the residual stress at the surface of the workpiece [15]. In further works, a varied thermal effect was related to the energy partition [16], the contact time [17], and the temperature [18] during different grinding experiments.

In addition to experimental research, numerical methods have also been developed. Holtermann numerically determines the thermo-mechanical loads of the workpiece for traverse grinding [19]. Pashnyov describes a mathematical model of stress state based on grinding forces [20]. Thereupon the model was extended to evaluate the temperature distribution in a three-layer metal composite [21].

However, a controlled generation of surface layer properties for machining processes such as grinding is hardly possible yet [22].

The reason for this is the mainly process-parameter-oriented view on the thermal effect prevailing in many scientific works, in which correlations between process parameters and material modifications are investigated. The Collaborative Research Center (CRC) SFB/TRR 136 takes a different approach by introducing the concept of Process Signatures in which material modifications are correlated with internal material loads generated during the manufacturing process [23]. In the context of this material-oriented consideration, it is assumed that the cause-effect relationships between the quantities shown in Figure 1 are most relevant for grinding.

In order to predict the material change due to grinding, it is necessary to understand these cause-effect relationships during the process. For grinding as a thermo-mechanical process, the stresses, strains, and temperatures of the material, as well as their local and temporal gradients, can be considered as the main internal material loads leading to material modifications. In this paper, cause-effect relations between internal material load quantities and material modifications due to thermal effects are investigated, which correspond to correlation 3, the Process Signature. It consists of different components showing a specific correlation, e.g., between a material modification, such as a hardness change or a residual stress change, and an internal material load, such as a local temporal temperature gradient.

So far, many studies have investigated the thermal impact of grinding on the workpiece and have shown the effects of process parameters on process quantities (forces, power) and on surface integrity. Temperatures were also considered in this context. To date, however, there are no clear functional dependencies between the intensity and development of internal material loads and the resulting material modifications in terms of the extent of modification and depth effect. This paper aims at contributing to close this knowledge gap and to identify correlations between the internal material loads due to thermal impact and the resulting modifications.

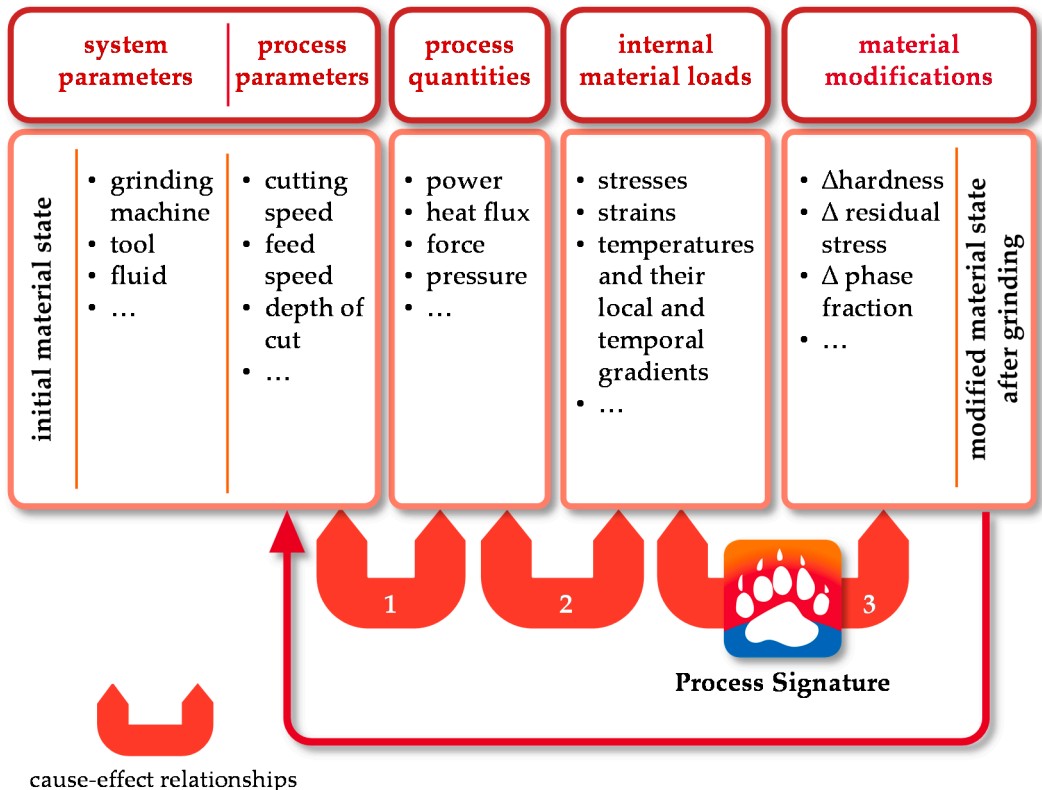

**Figure 1.** Cause-effect relationships according to the Collaborative Research Center SFB/TRR 136 "Process Signatures".

A nomenclature describing all symbols used in the paper is shown in Table 1.

**Table 1.** Nomenclature.

| Symbol | Description | Unit |
|---|---|---|
| $\dot{q}$ | heat flux density | W/mm$^2$ |
| $\partial T/\partial t$ | temperature rate | °C/s |
| $a_e$ | depth of cut | μm |
| $a_{ed}$ | depth of cut (dressing) | μm |
| $b_s$ | grinding wheel width | mm |
| $d_s$ | grinding wheel diameter | mm |
| $d_t$ | diameter thermocouple | mm |
| FEM | finite element method | - |
| $h_w$ | workpiece height | mm |
| $l_g$ | contact length | mm |
| $l_w$ | workpiece length | mm |
| mcs | metallographic cross-sections | - |
| MWF | metalworking fluid | - |
| $P_c''$ | specific grinding power | W/mm$^2$ |
| $P_{HJ}$ | Hollomon–Jaffe parameter | - |
| $Q_{MWF}$ | flow rate of the fluid | L/min |
| $R^2$ | coefficient of determination | - |
| rs | residual stress | MPa |
| T | temperature | °C |
| t | time | s |
| $t_{0.9}$ | response time (thermocouple) | s |
| $t_c$ | contact time | s |

**Table 1.** *Cont.*

| Symbol | Description | Unit |
|---|---|---|
| $U_d$ | overlapping ratio | - |
| $v_{ft}$ | tangential feed speed | m/min |
| vh | Vickers hardness | HV1 |
| $v_s$ | grinding wheel speed | m/s |
| $v_{sd}$ | grinding wheel speed (dressing) | m/s |
| $w_v$ | workpiece width | mm |
| z | depth below surface | mm |
| $\alpha$ | heat transfer coefficient | $W/(m^2 \cdot K)$ |
| $\sigma$ | residual stress | MPa |

## 2. Objectives and Research Approach

The presented paper aims at determining a functional relationship between material modifications and thermal material loads during grinding by using the concept of Process Signatures. Figure 2 shows the procedure to achieve this goal.

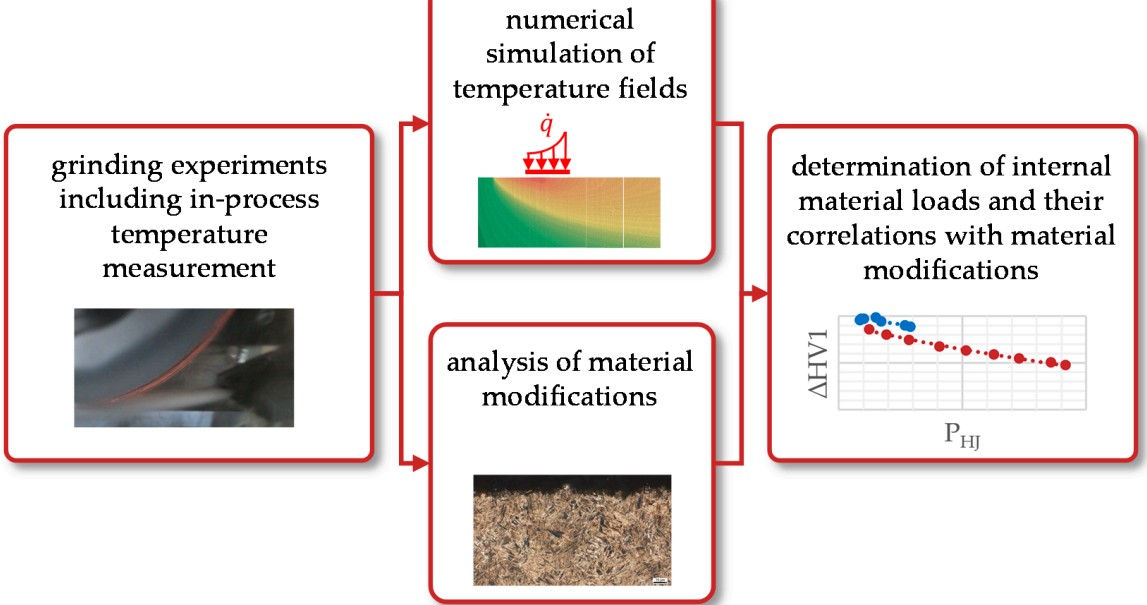

**Figure 2.** Procedure and expected outcome.

The focus of this work is on thermal loads and their effect on the hardness change at different depths below the surface and on the residual stress change at the surface for grinding. These modifications are correlated to the maximum temperature, the maximum temperature gradient, and the contact time $t_c$, which can be considered to be the most important characteristics for heat treatment processes without phase transformation [24]. Furthermore, the material modifications are correlated with maximum temperature rates and the Hollomon–Jaffe parameter $P_{HJ}$ as they can be characteristics for the annealing of the surface and subsurface layer and therefore for the hardness change of the material [25]. The maximum temperature, maximum temperature gradient, and maximum temperature rate were determined using temperature fields generated numerically by modeling a heat source moving parallel to the workpiece surface in the direction of the tangential feed speed $v_{ft}$.

## 3. Materials and Methods

The initial length, width, and height of the prismatic workpieces used in the experimental investigation were $l_w$ = 150 mm, $w_w$ = 18 mm, and $h_w$ = 35 mm. The workpiece material was an AISI 4140 (42CrMo4) steel homogeneously austenitized at 850 °C for 2 h and quenched in oil. The chemical composition of this steel is shown in Table 2. After this heat treatment, the hardness of the workpieces was 670 HV1. Thermocouples of type K (diameter $d_t$ = 0.25 mm, response time $t_{0.9}$ < 0.15 s) were fixed in holes in the lateral flat side surface of the workpieces. The measuring positions of the thermocouples were at half-width of the workpieces (9 mm) in three different depths $z_1$ = 1 mm, $z_2$ = 2 mm, and $z_3$ = 3 mm below the initial surface. No further processing of the workpieces was carried out before the grinding experiments. Thus, the material modifications investigated in this paper are resulting from the surface grinding processes only.

**Table 2.** Chemical composition of AISI 4140 (42CrMo4).

| AISI 4140 Chemical Composition (%) | | | | | | | |
|---|---|---|---|---|---|---|---|
| C | Si | Mn | P | S | Cr | Mo | Ni |
| 0.448 | 0.264 | 0.735 | 0.012 | 0.002 | 1.09 | 0.244 | 0.2 |

The grinding experiments were performed on a Blohm Profimat 412 HSG (Hamburg, Germany) profile grinding machine. The experimental setup is shown in Figure 3.

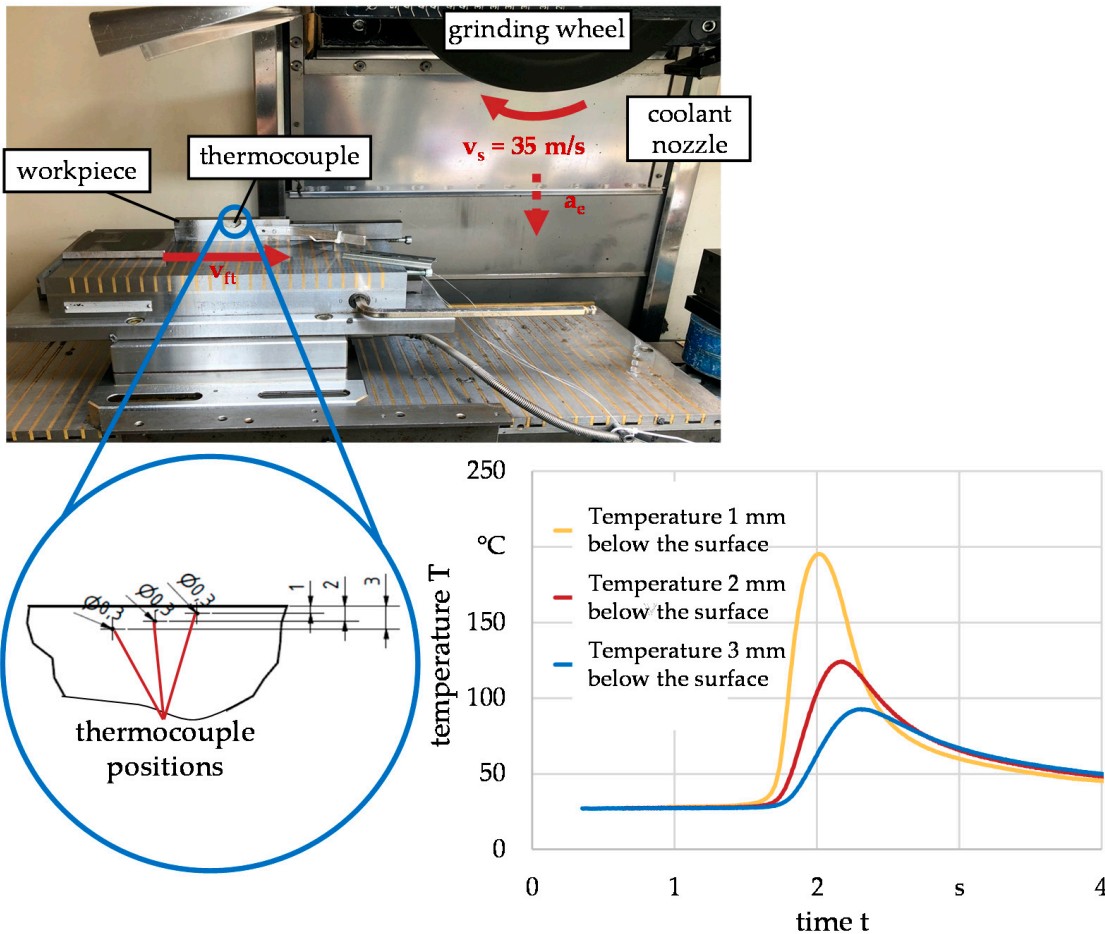

**Figure 3.** Experimental grinding setup in surface grinding including temperature measurements using thermocouples in different depths below the surface.

The process was carried out as single-step surface grinding with constant tangential feed speed $v_{ft} = 1$ m/min and grinding wheel speed $v_s = 35$ m/s without sparking out. The used grinding wheel was a vitrified bond corundum wheel (type A80HH8V, $d_s = 400$ mm, width $b_s = 20$ mm). Before each grinding test, the grinding wheel was cleaned using a cleaning nozzle in order to prevent clogging effects and dressed with a single-point diamond dresser to generate similar conditions regarding the microtopography on the surface of the tool. The overlapping ratio $U_d = 3$, depth of cut $a_{ed} = 30$ μm (three times) and grinding wheel speed $v_{sd} = 35$ m/s during dressing were kept constant. The grinding fluid was a universal grinding oil. The flow rate of the fluid $Q_{MWF}$, as well as the depth of cut $a_e$, are varied in the grinding tests to generate different thermal impacts on the workpiece, as shown in Figure 4. The material analysis included measurements of the residual stress depth profiles and the analysis of the metallographic cross-sections for all grinding conditions. For selected grinding conditions, hardness depth profiles were measured as well. For residual stress measurement, an X-ray diffractometer of type MZ IV (GE Inspection Technologies, Ahrensburg, Germany) equipped with a position-sensitive detector was used with vanadium-filtered Cr-K$\alpha$ radiation.

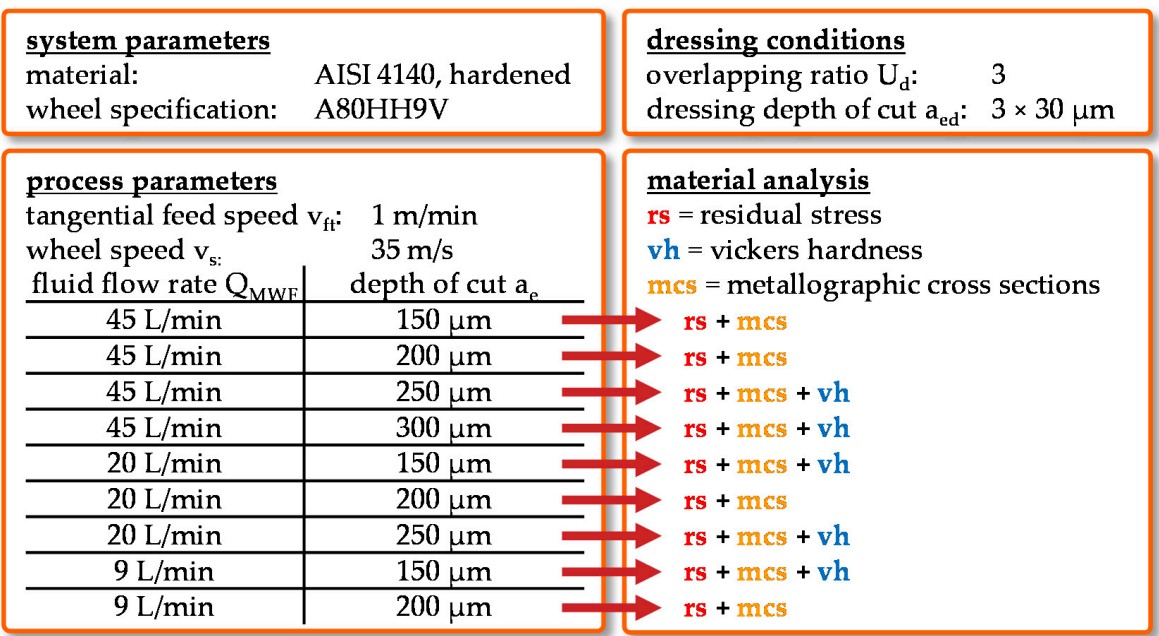

**Figure 4.** Design of experiments.

The simulations are carried out on a Linux server (1 AMD Epyc 7452 with 32 Cores/ 64 Threads, 256 GB RAM). The 3D simulations with the finite element code SYSWELD were carried out for sample length of 50 mm, width of 15 mm, and height of 18 mm. The minimal distance between two Gauss points was 40 μm in the surface near the area. In that region, the largest gradients occur. The complete geometry was modeled with 34,000 quadrangle volume elements with 8 Gauss points each. Because of symmetry conditions, only one-half of the workpiece was considered. The simulations neglected any material loss due to material removal. The wheels material was not considered. Within the contact zone between the workpiece and grinding wheel, the heat impact $\dot{q}$ is modeled by means of a cubic function for a better approximation [26]. Each temperature simulation lasted approximately 3 h. The FE tool was used in order to calculate temperature profiles and derive temperatures, temperature gradients, and rates in different depths below the surface as well as at the surface. The thermophysical properties of the workpiece, which are presented in Table 3, were used according to Richter [27].

**Table 3.** Thermophysical properties used in SYSWELD.

| Thermophysical Properties | Temperature, °C | | | | | | | | | | |
|---|---|---|---|---|---|---|---|---|---|---|---|
| | 0 | 20 | 100 | 200 | 300 | 400 | 500 | 600 | 700 | 800 | 900 |
| Thermal conductivity λ, W/(m·K) | 36.8 | 37.0 | 37.6 | 37.8 | 36.2 | 34.8 | 33.2 | 31.1 | 29.1 | 20.5 | 11.7 |
| Specific heat capacity $c_P$, J/(kg·K) | 435 | 446 | 516 | 528 | 539 | 550 | 562 | 573 | 584 | 596 | 607 |
| Density ρ, kg/m³ | 8039 | 8029 | 7989 | 7938 | 7889 | 7840 | 7792 | 7744 | 7697 | 7650 | 7604 |

The model is based on a heat source representing the grinding wheel moving parallel to the workpiece surface. The procedure for determining the temperature profiles is illustrated in Figure 5.

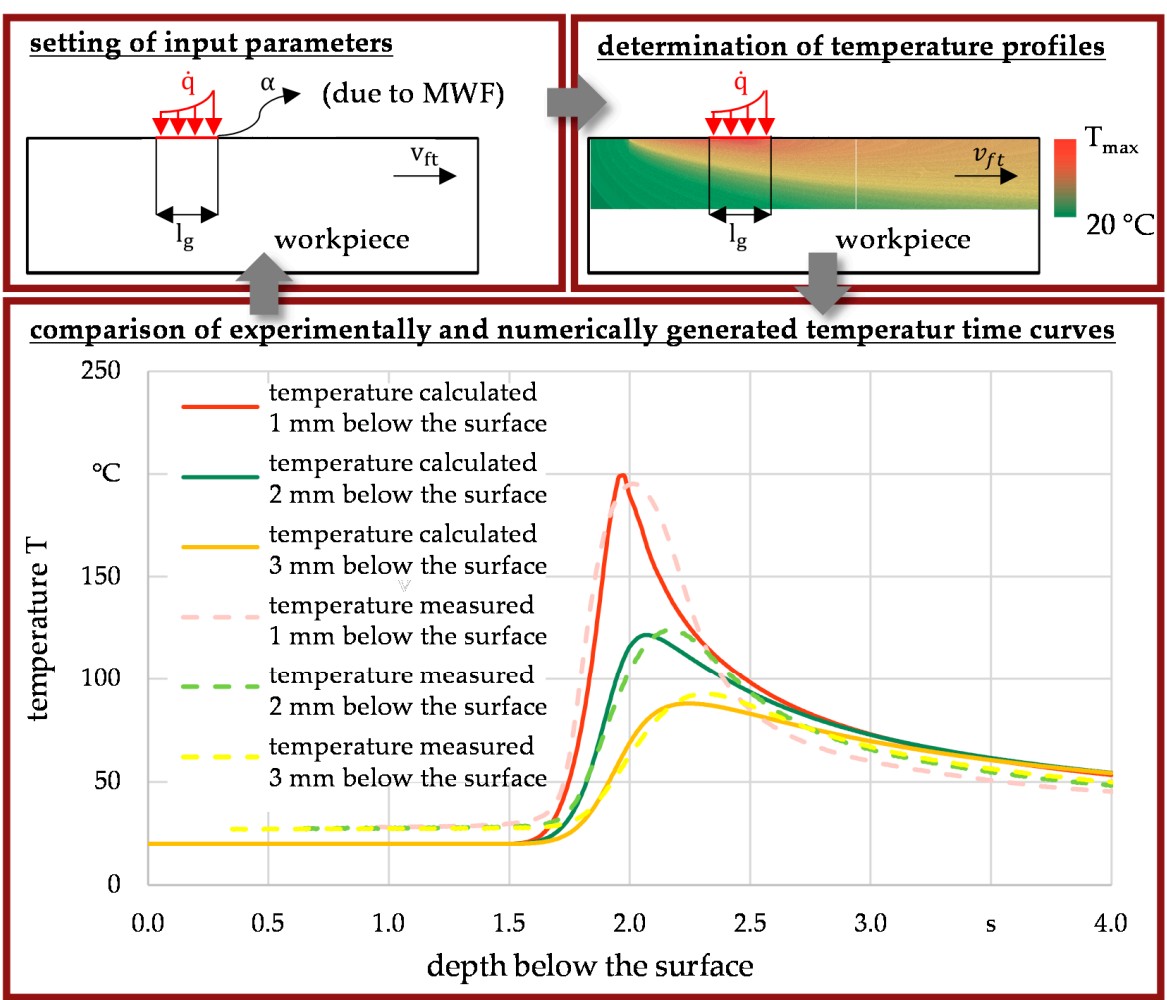

**Figure 5.** Procedure for validating the numerically generated data.

The heat flux density and heat transfer coefficient were at first roughly estimated from power and process force measurements and also by data from literature [28,29] for every grinding setup. In additional steps, these quantities were adjusted and fine-tuned in the FE model in order to compensate deviations between experimentally measured and calculated temperature-time curves in depths of 1, 2, and 3 mm below the initial surface. The numerically determined temperature-time curves were consistent with the experimental data for heat flux densities of $\dot{q}$ = 6.5–17.7 W/mm² and heat transfer coefficients of $\alpha$ = 5000–60,000 W/(mm²·K), which are varied depending on the grinding

process considered in the FE model. The temperature profiles were then used to determine the investigated thermal loads close to the workpiece surface. By this procedure, the numerically calculated temperature profiles next to the surface were calibrated by the measured temperatures several millimeters below the surface.

## 4. Experimental and Numerical Findings

The ground workpieces were analyzed in terms of the residual stress (by X-ray diffraction), the hardness (by Vickers hardness measurement), and the microstructure (by metallographic cross-section analysis). Presented in Figure 6 are the residual stress depth profiles for every grinding parameter combination. Based on the experience from preliminary investigations, it can be assumed that the residual stress before grinding was approximately $\sigma_{II} = 0$ MPa.

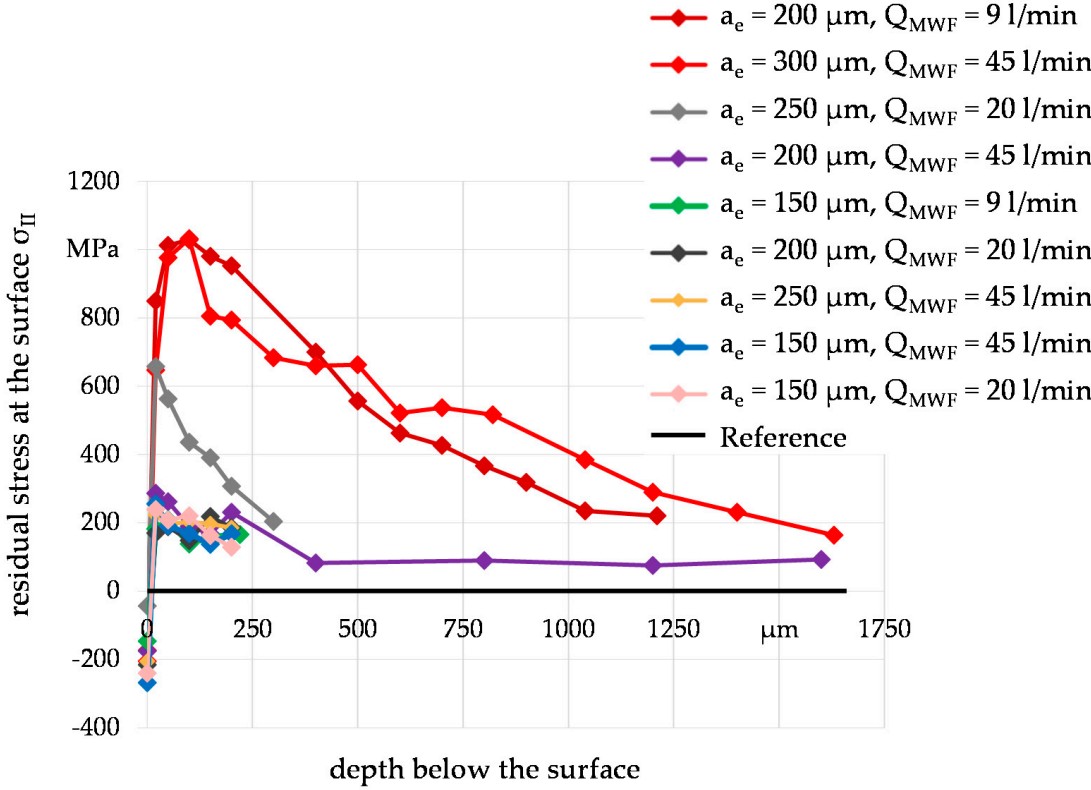

**Figure 6.** Residual stress depth profiles after grinding for different process parameter combinations.

The residual stresses are measured in depths of 0, 20, 50, 100, 150, and 200 µm below the surface for every grinding experiment. For some workpieces ground under specific grinding parameters, the residual stresses are measured deeper below the surface to provide an indication of the depth range of the generated modification by grinding. It can be concluded that three of nine investigated grinding setups lead to noticeable deep ranges of the residual stress, which were (i) $a_e = 200$ µm, $Q_{MWF} = 9$ L/min, (ii) $a_e = 300$ µm, $Q_{MWF} = 45$ L/min and (iii) $a_e = 250$ µm, $Q_{MWF} = 20$ L/min. The corresponding high-tensile residual stresses in a depth of 250 µm up to more than 1 mm below the ground surface are typical for grinding processes with a high thermal impact. Considering the depth profiles of workpieces ground by the other process parameters, only marginal differences occur. It can be assumed that the depth effect of the thermal load influencing the residual stress was similar and did not exceed a depth of 20 µm below the surface. Nevertheless, the influence of the thermal load on the residual stresses at the surface is also worth mentioning. The change of the residual stresses depending on the thermal load is discussed in more detail in Section 4.2.

The Vickers hardness measurements HV1 were taken up to a depth of 2 mm below the surface in steps of 50 μm. The results are shown in Figure 7. In order to maintain clarity, not all grinding conditions were examined. From the residual stress depth curves, it can be derived that the hardness depth curves of the omitted conditions show a similar trend to the trends shown in Figure 7.

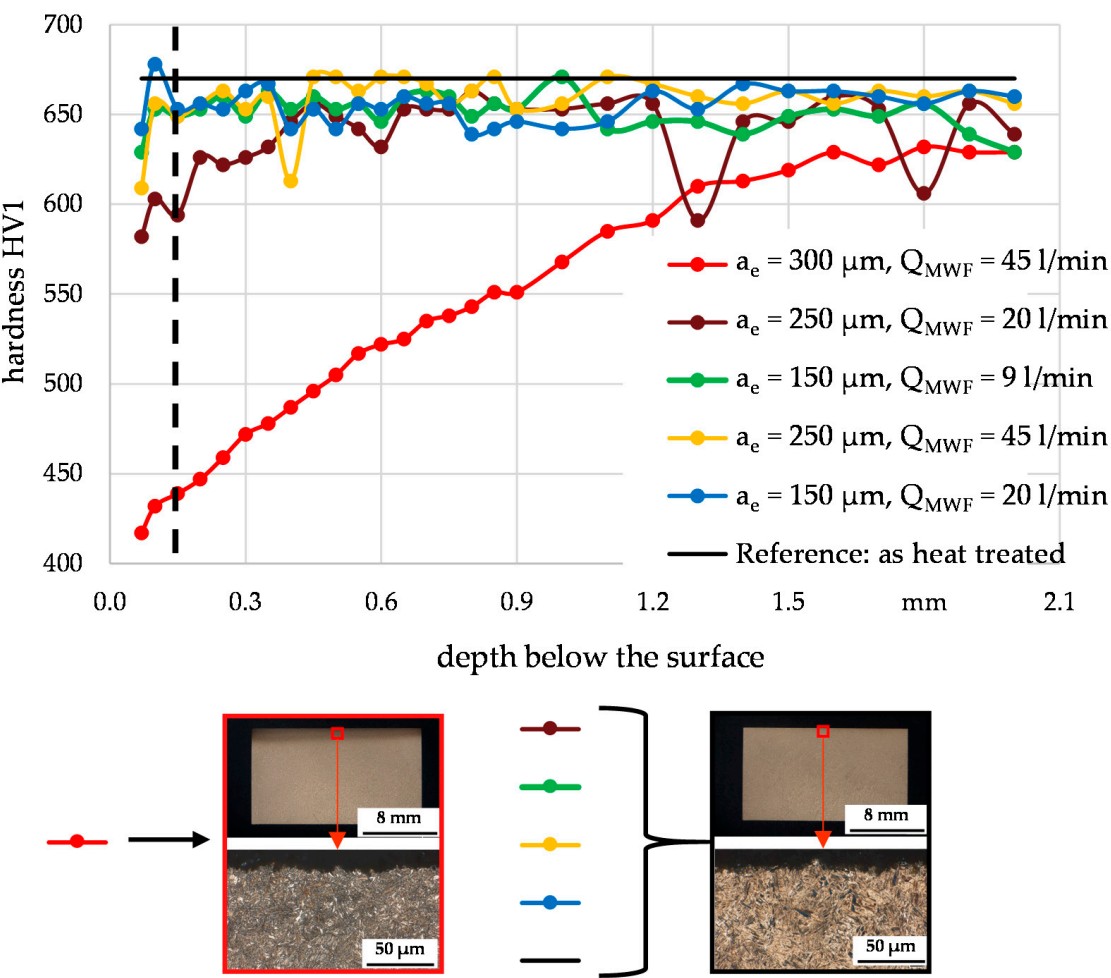

**Figure 7.** Hardness depth profiles after grinding for different process parameter combinations.

As for the residual stress depth profile, the hardness of the workpiece ground with a depth of cut of $a_e = 300$ μm and a flow rate of $Q_{MWF} = 45$ L/min exhibited a significantly larger depth range than the workpieces ground under other conditions. The indicated annealing effects in the metallographic cross-sections lead to the result that this was due to the thermal effect, which is consistent with the corresponding high-tensile residual stresses. The cross-section analysis also showed that no phase transformation occurred during the grinding experiments. A likewise higher depth range of the residual stress and the hardness can be observed when grinding with a depth of cut of $a_e = 250$ μm and $Q_{MWF} = 20$ L/min, which is about 350 μm. Although no tempering zones are observed via metallographic cross-section analysis, it can be assumed that grinding under these conditions initiates a change in mechanisms that results in hardness changes. The vertical dashed line indicates the depth from which a reliable measurement of the hardness via the Vickers method is possible. Therefore measured hardness values next to the surface were not considered in the following evaluation of the hardness changes due to thermal loads.

It is generally known that high thermal loads generated during grinding can lead to a decrease in hardness and a formation of tensile residual stresses, which can also be derived from the hardness and residual stress depth profiles measured after grinding with a depth

of cut of $a_e$ = 300 µm and a flow rate of $Q_{MWF}$ = 45 L/min in Figures 6 and 7. Nevertheless, it is still not clear which material internal load quantity or quantities is/are most appropriate to describe certain material modifications for grinding. Different approaches discussing correlations between the hardness change and residual stress change at the surface and thermal loads are presented in Sections 4.1 and 4.2, respectively. Additionally, functional relationships are provided to describe the hardness and residual stress change dependent on specific thermal loads.

### 4.1. Evaluation of Hardness Changes Due to Thermal Loads

In a first approach, the correlation between the hardness change and the maximum temperature was investigated, which is illustrated in Figure 8. The shown values represent the hardness changes in different depths below the surface and the numerically calculated maximum temperatures at the respective depths.

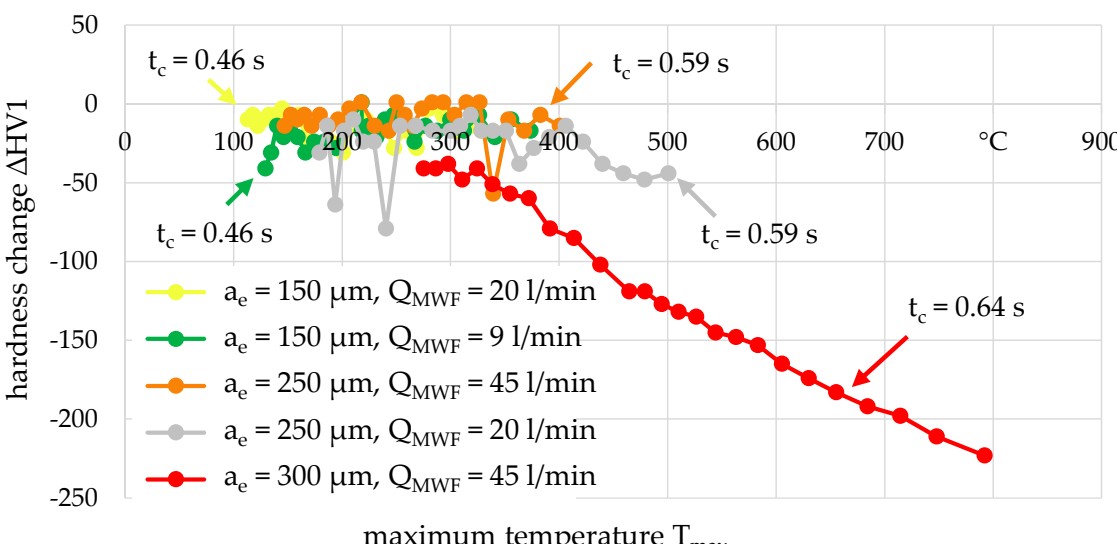

**Figure 8.** Hardness change dependent on the maximum temperature for different process parameter combinations.

In general, the maximum temperatures decrease with a greater distance from the surface, which can be observed when applying the process parameters marked by the red and gray dots. This corresponds to the expectations. As can already be expected from the hardness and residual stress depth profiles, the process parameters marked by the red dots correspond to considerable higher maximum temperatures close to $T_{max}$ = 800 °C. Reasons for this could be the highest depth of cut ($a_e$ = 300 µm) together with the highest contact time ($t_c$ = 0.64 s) among the applied process parameter combinations. No significant changes can be observed considering the other process parameter combinations. The presented findings cannot be attributed exclusively to the maximum temperature, as different process parameter combinations led to equal maximum temperatures but different hardness changes.

In a second approach, it is examined if the hardness change can be appropriately described as a function of the maximum temperature gradient $(\partial T/\partial z)_{max}$, which is the local gradient perpendicular to the surface. The relationships for the different process parameters investigated are shown in Figure 9.

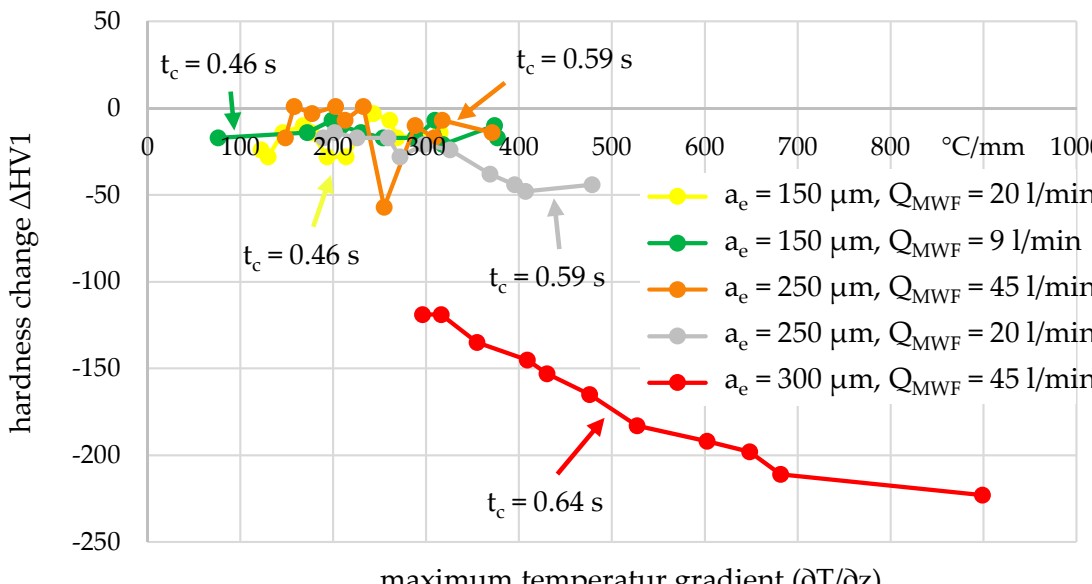

**Figure 9.** Hardness change dependent on the maximum temperature gradient for different process parameter combinations.

Comparing Figures 8 and 9, similar trends can be observed showing significant differences in the hardness changes after grinding with a depth of cut of $a_e$ = 300 μm and a flow rate of $Q_{MWF}$ = 45 L/min for a given maximum temperature (Figure 8) or temperature gradient, respectively (Figure 9). These figures clearly show that both the maximum temperature as well as the maximum temperature gradient are, at least, not exclusively responsible for the hardness change in grinding.

From literature, it is known that also a time-dependent quantity has to be taken into account, considering the softening effect due to thermal loads. The reason for this is the recovery process, e.g., the annihilation of dislocations, leading to the softening of the material. The more time for the recovery process is available, the greater is the reduction in hardness for a given temperature. However, the determination of the exact time the temperature affects the workpiece material during grinding is complicated. A quantity for a rough estimation of the time of temperature impact on the workpiece surface and subsurface is the contact time $t_c$, which in surface grinding corresponds to the geometric contact length $l_g$ divided by the tangential feed speed $v_{ft}$ [12,17,30]. In accordance with investigations by Takazawa [30], Figure 10 shows the observed hardness reduction as a function of the contact time for constant maximum temperatures.

It can be observed that for a given contact time, a decreasing maximum temperature leads to a decrease in the hardness reduction. For a given maximum temperature, a decrease in the contact time from $t_c$ = 0.64 s to $t_c$ = 0.59 s resulted in a hardness reduction that could only be plotted for maximum temperatures $T_{max} \leq 500$ °C. Higher maximum temperatures were only determined for a contact time of $t_c$ = 0.64 s. Both results were in line with the previous results. Nevertheless, for a given temperature of $T_{max}$ = 350 °C and $T_{max}$ = 390 °C, slightly different hardness changes of $\Delta HV1$ = 7–14 occurred, although both the maximum temperature and the contact time were constant in each case. Due to this, another quantity is also investigated to describe the time-dependent thermal effect more specifically.

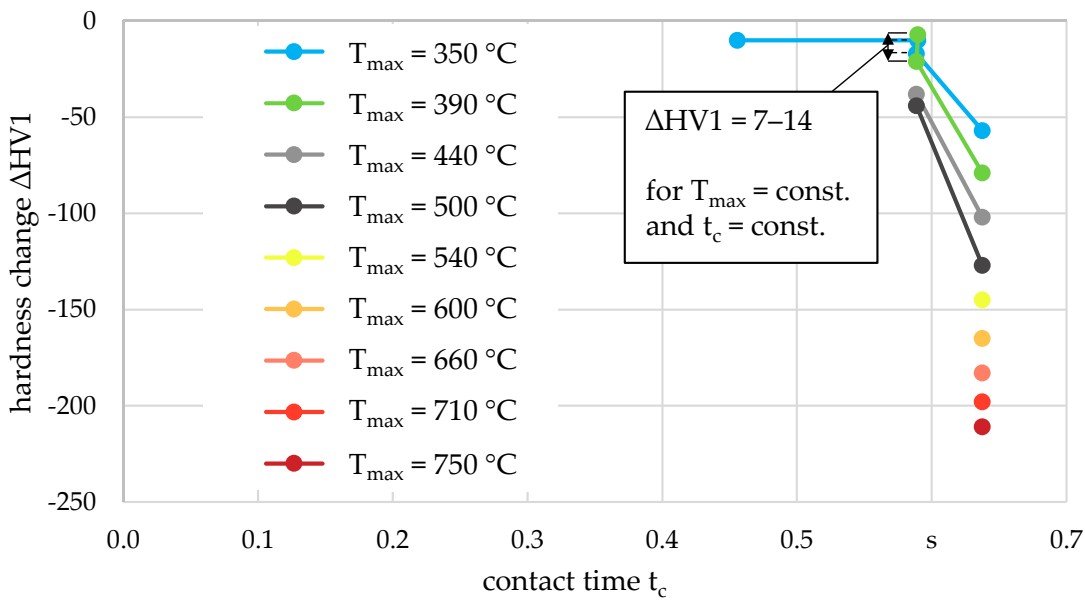

**Figure 10.** Hardness change dependent on the contact time for given maximum temperatures.

One quantity that can be directly related to the degree of the recovery process leading to softening of the workpiece material is the maximum temperature rate. For various process parameter combinations, the maximum temperature rate at various depths below the surface is determined below. More specifically, the maximum temperature rates were taken from the temperature-time curve in the heating phase, which is in the region of the inflection point of the curve. The change in hardness as a function of the maximum temperature rate is shown in Figure 11.

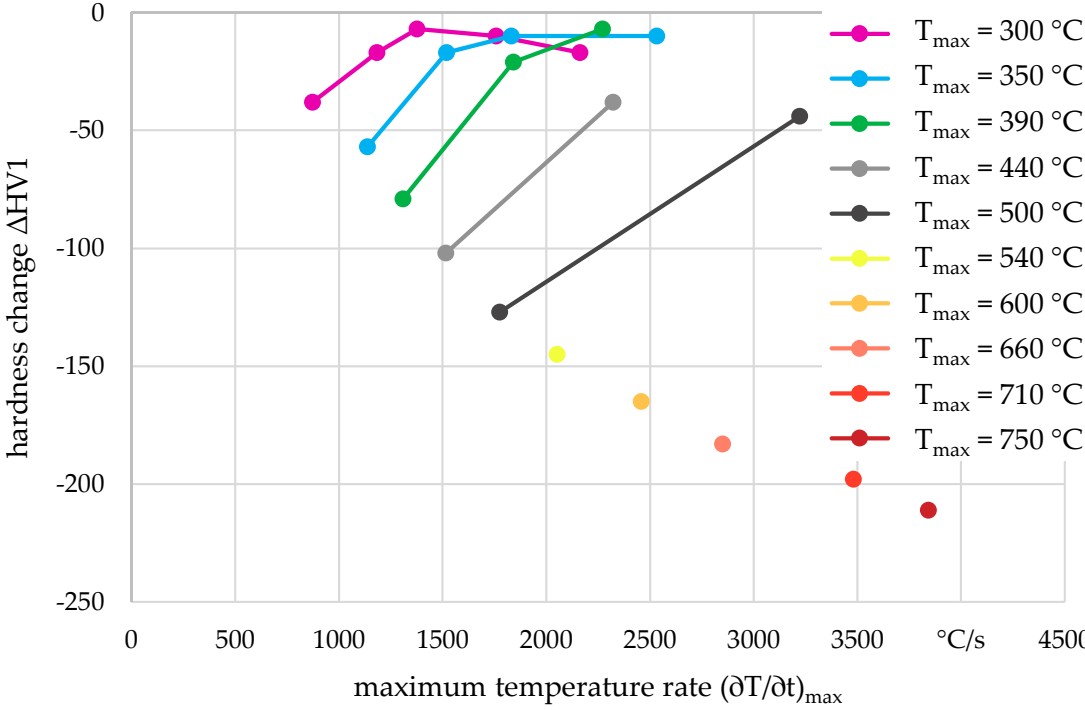

**Figure 11.** Hardness change dependent on the maximum temperature rate for given maximum temperatures.

For the given maximum temperatures, an increase in the maximum temperature rate resulted in a decreasing reduction in the hardness change. This can be explained

by the fact that a higher maximum temperature rate led to less time being available for the softening of the material. Considering the hardness change development for given maximum temperatures of $T_{max}$ = 350 °C and $T_{max}$ = 390 °C, it can be observed that even slightly different changes of the hardness can be attributed to a specific maximum temperature rate. In contrast to that, different hardness changes were measured for constant contact times and maximum temperatures. Comparing the contact time and the maximum temperature rate, it can be concluded that the maximum temperature rate provides a more precise description of the temporal effect of the temperature in grinding.

Functional relationships of hardness change and maximum temperature rates are limited to both the maximum temperature and the range of maximum temperature rates determined for a given maximum temperature. For the given maximum temperatures of $T_{max}$ = 350 °C and $T_{max}$ = 390 °C, it can be assumed that at a maximum temperature rate of about $(\partial T/\partial t)_{max}$ > 2300 °C/s, the measured hardness is nearly the same as before grinding. This can be explained by the short time that the temperature is present to influence the recovery process associated with the softening of the material. If higher maximum temperatures are present, a softening effect occurs even at higher temperature rates and, consequently, at shorter heating times. The changes in hardness for maximum temperatures of $T_{max}$ = 440 °C and $T_{max}$ = 500 °C indicate a similar trend to that for lower maximum temperatures. Nevertheless, more data points are required to improve the reliability of the trends for maximum temperatures of $T_{max}$ = 440 °C and $T_{max}$ = 500 °C and to generate trends for maximum temperatures of $T_{max} \geq$ 540 °C.

The approach shown in Figure 11 implies that two quantities of the thermal load were required to describe the hardness change, which was the maximum temperature rate and the maximum temperature. Thus, each given maximum temperature led to a specific function. To reduce a large number of possible functional relationships, another approach is investigated, which is illustrated in Figure 12. The same data as in Figure 11 were used, neglecting the maximum temperatures and taking into account the metallographic cross-section analyses. The measured hardness changes were then separated into two categories distinguishing whether in the metallographic cross-section analyses a heat-affected zone is observed or not, shown exemplarily in Figure 12.

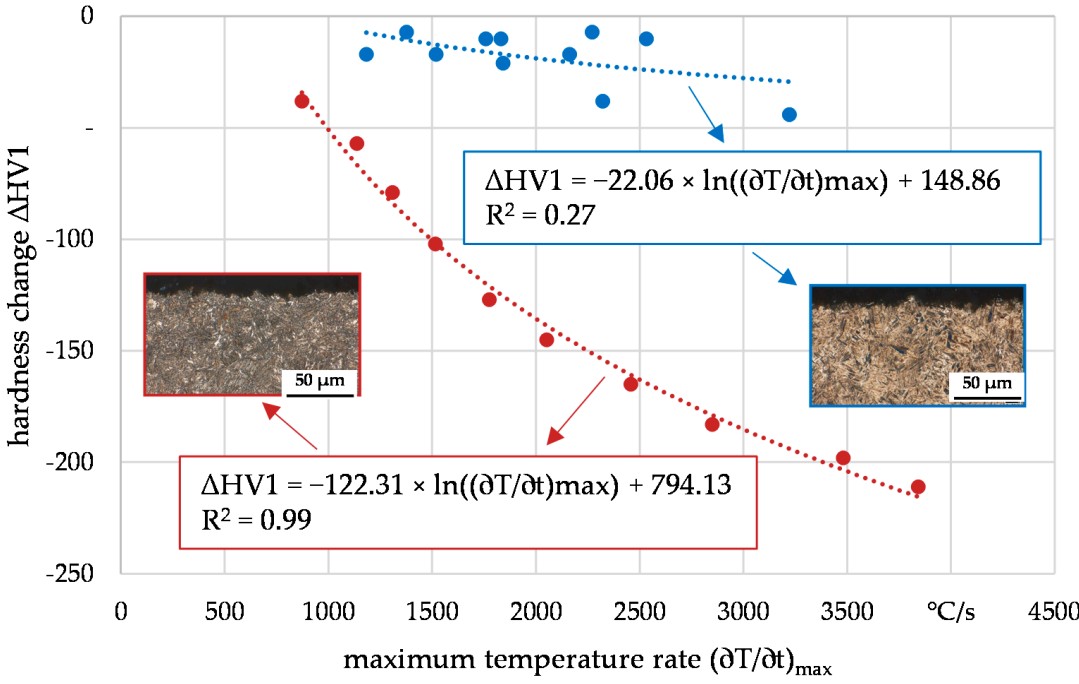

**Figure 12.** Hardness change dependent on the maximum temperature rate in consideration of metallographic cross-section analysis (blue: no visible heat-affected zone; red: visible heat-affected zone).

A regression curve was determined for the red doted values (heat-affected zone was observed), and the blue dotes values (no heat-affected zone was observed), respectively. For the regression curve assigned to the values where a heat-affected zone was observed, a high coefficient of determination ($R^2$ = 0.99) is calculated, whereas, for the other regression curve that implies cross-section analyses where no heat-affected zone was observed, a low coefficient of determination ($R^2$ = 0.27) is shown. It can be concluded that the introduced categories depending on the results of the metallographic cross-section analyses helped to reduce the complexity of the functional description of the hardness change compared to the approach shown in Figure 11. Nevertheless, the low coefficient of determination for the blue regression curve suggests that another quantity may be more appropriate to describe the hardness change occurring when no heat-affected zone can be observed.

As a final approach, the correlation between the hardness change due to grinding and the Hollomon–Jaffe parameter $P_{HJ}$ is investigated. In the classic Hollomon–Jaffe parameter, the maximum temperature and the time when the temperature is present are included, which is why this quantity could also be evaluated as an internal material load. Usually, the Hollomon–Jaffe parameter is used for classic heat treatment processes describing the tempering of steels by $P_{HJ} = T(C + \log t)$ [31]. In this context, T denotes the absolute temperature of the tempering process, C is a constant characteristic of the tempered steel, and t is the tempering time. For grinding, this parameter was used to estimate changes at the surface and subsurface [32]. However, the estimation of the annealing time is usually difficult. Kaiser et al. showed that the hardness of hardened AISI 4140 steel correlated with the Hollomon–Jaffe parameter for different heating rates up to 1200 °C/s, using another approach to calculate the parameter [25]. In their work, they used the incremental description of the Hollomon–Jaffe parameter, which is applicable for non-isothermal heating, described by Gomes et al. [33] and shown in Equation (1):

$$P_{HJ,i} = P_{HJ,i-1} + \frac{T_i}{2.303 \cdot 10^{\frac{P_{HJ,i-1}}{T_i} - C}} \cdot \Delta t \tag{1}$$

The same approach is used by Balart et al. to characterize the softening behavior for grinding different steels [14]. A description of the functional relationship between the hardness change and the Hollomon–Jaffe parameter in grinding has not been carried out yet.

In the present paper, this approach is used to provide a more precise description of the absolute temperature and temporal development than the maximum temperature and the maximum temperature rate in grinding, exemplified in Figure 13.

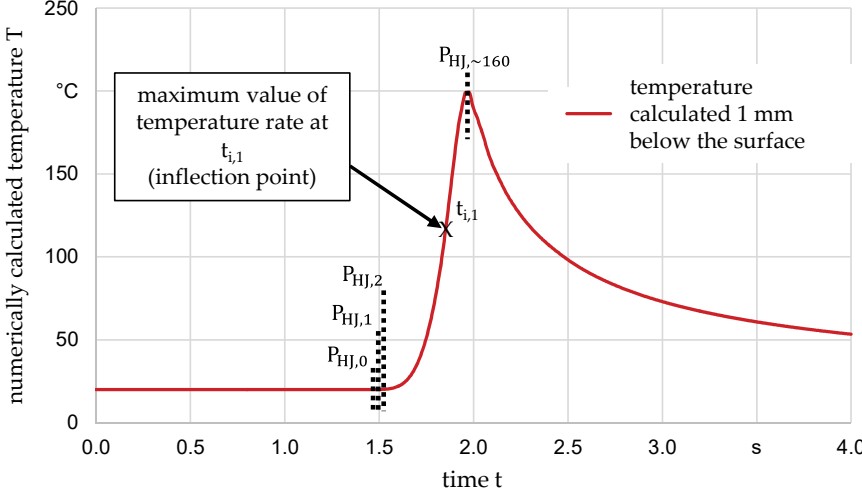

**Figure 13.** Incremental calculation of the Hollomon–Jaffe parameter $P_{HJ}$ depending on the time temperature development.

In the incremental description of the Hollomon–Jaffe parameter, the temperature increase and the time in which this increase occurs are taken into account in each incremental time interval, whereas the maximum temperature rate only represents the temporal behavior at the inflection point of the temperature-time curve. As the initial condition, the Hollomon–Jaffe parameter is set at $P_{HJ,0} = C \times T_0$ with the constant characteristic of the material being $C = 18.69$ [31] and the starting temperature $T_0 = 294.15$ K, which corresponded to the ambient temperature. Each time increment of $\Delta t = 0.004$ s, the Hollomon–Jaffe parameter $P_{HJ,I}$ is calculated according to Equation (1) until the maximum temperature is reached, being at $P_{HJ,160}$ in the example shown in Figure 13.

The hardness change as a function of the incrementally calculated Hollomon–Jaffe parameter is shown in Figure 14.

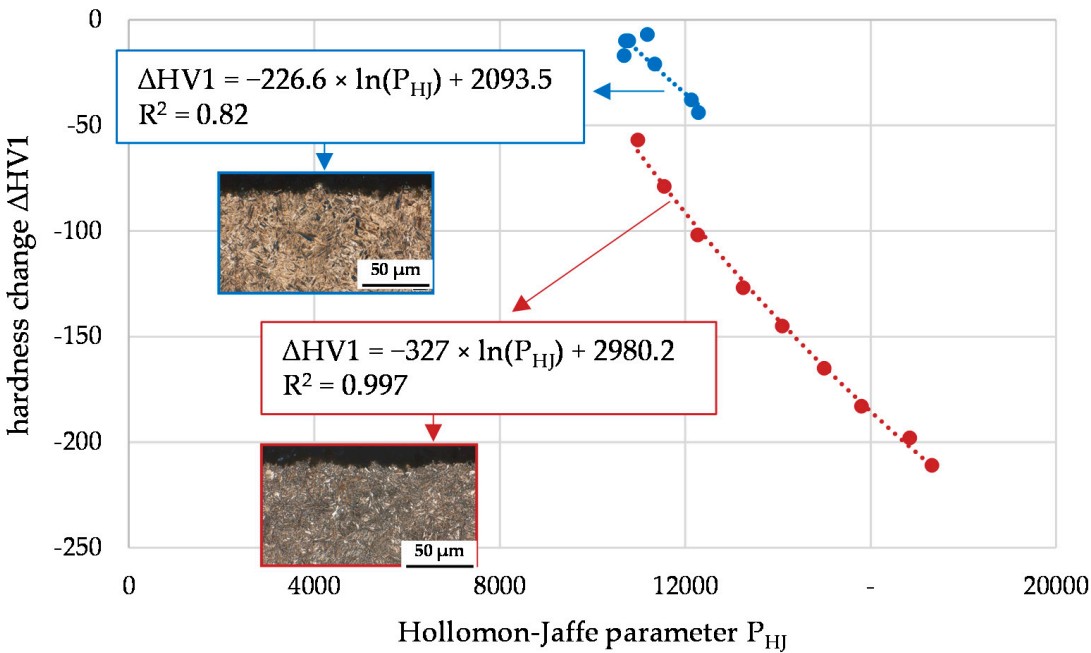

**Figure 14.** Hardness change dependent on the Hollomon–Jaffe parameter in consideration of metallographic cross-section analysis (blue: no visible heat-affected zone; red: visible heat-affected zone).

Considering the cross-section analyses, regressions with high coefficients of determination were determined for both workpiece areas where heat-affected zones were detected (red line, $R^2 = 0.997$), but also for those where no heat-affected zone was observed (blue line, $R^2 = 0.82$). Due to this, the Hollomon–Jaffe parameter according to (1) can be considered the most appropriate among the investigated internal material load quantities to describe the hardness change due to the thermal effect in grinding.

### 4.2. Evaluation of Residual Stress Changes Due to Thermal Loads

In this section, correlations between the residual stress change at the surface and different thermal load quantities were investigated to identify an appropriate thermal load quantity, describing its impact on the residual stress state at the surface. In contrast to the evaluation of the hardness changes, where these changes and the corresponding thermal loads were considered in different depths below the surface, there only is one value of the residual stress at the surface per process combination.

Comparable to Figure 12, the residuals stress change at the surface is correlated with the maximum temperature rate first. The result, also including the metallographic cross-section analyses, is shown in Figure 15.

For processes in which no heat-affected zone could be observed, the coefficient of determination of the regression line is $R^2 = 0.3$, which indicates that the maximum temper-

ature rate is not qualified for the description of the residual stress change. Two values (red dots in Figure 15) do not fit in the regression line, which can be a result of a change in the microstructure that is indicated by the metallographic cross-section analyses.

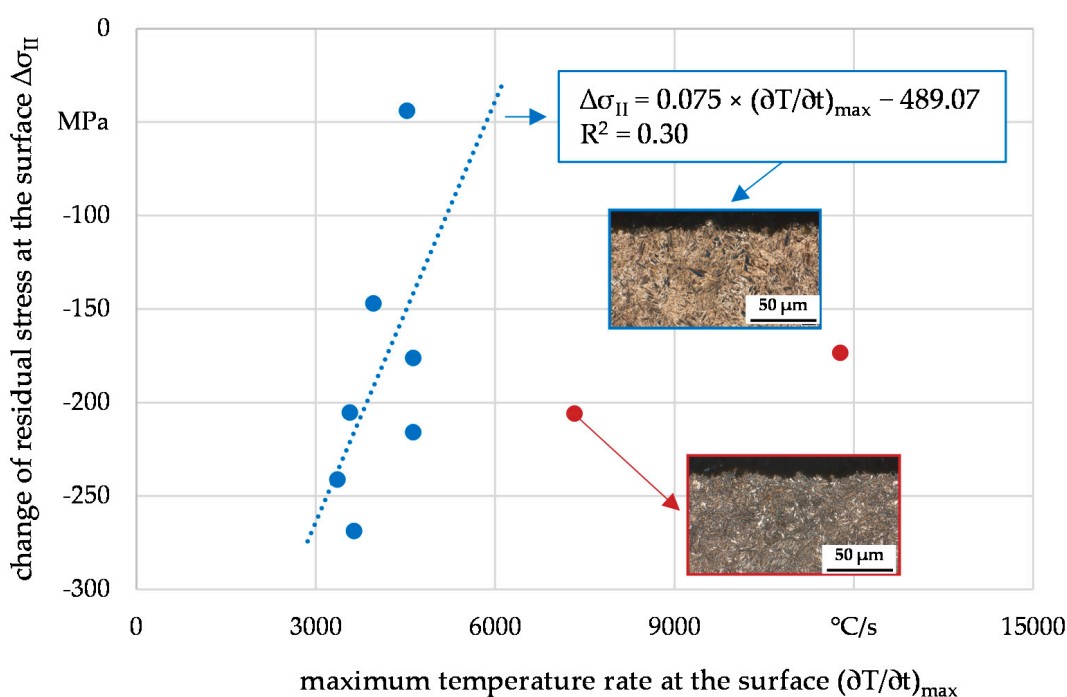

**Figure 15.** Change of residual stresses at the surface dependent on the maximum temperature rate in consideration of metallographic cross-section analysis (blue: no visible heat-affected zone; red: visible heat-affected zone).

As a second approach, the correlation between the residual stress change and the Hollomon–Jaffe parameter is investigated, which is illustrated in Figure 16. The Hollomon–Jaffe parameter is again determined according to Equation (1).

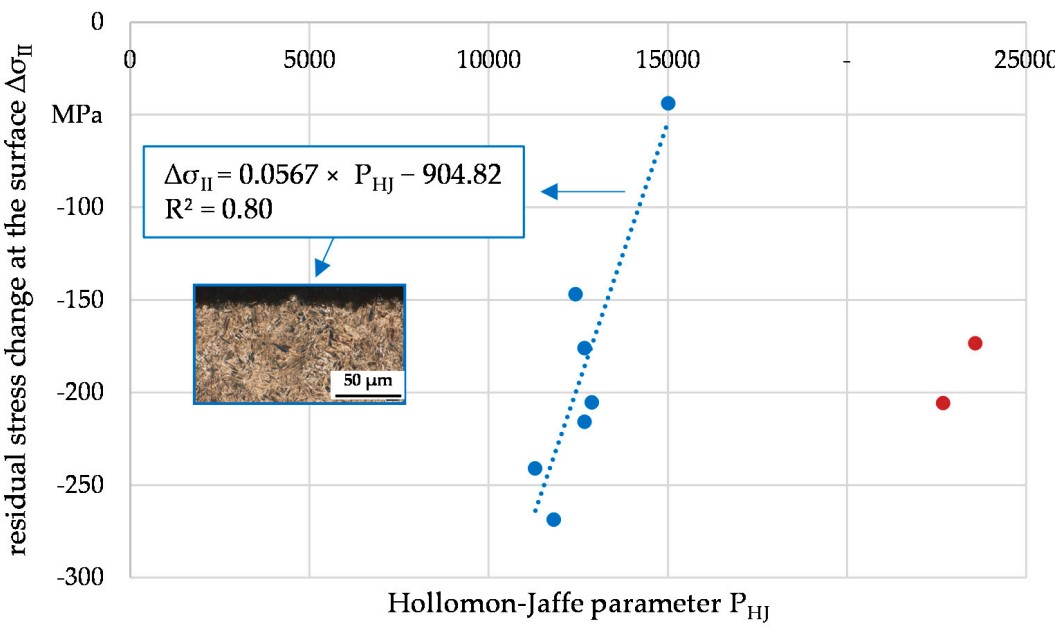

**Figure 16.** Change of residual stress at the surface dependent of the Hollomon–Jaffe parameter in consideration of metallographic cross-section analysis (blue: no visible heat-affected zone; red: visible heat-affected zone).

As with the change in hardness, the change in residual stress at the surface shows a good correlation with the Hollomon–Jaffe parameter. One reason for this could be that, in addition to the temporal development of the temperature, the absolute temperature is included in the calculation of the Hollomon–Jaffe parameter.

A more precise explanation for the formation of thermally induced residual stresses at the surface due to grinding is given by the temperature gradient [34]. The correlation of the change of the residual stress at the surface and the maximum temperature gradient is shown in Figure 17.

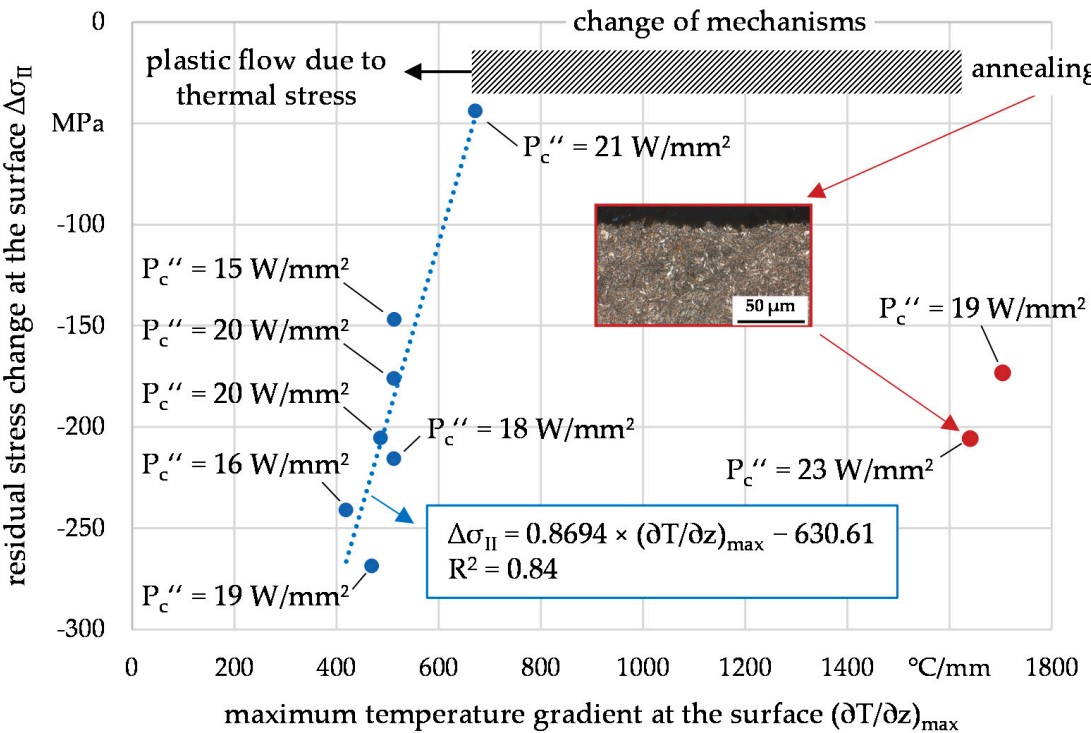

**Figure 17.** Change of residual stress at the surface dependent on the maximum temperature gradient at the surface in consideration of metallographic cross-section analysis (blue: no visible heat-affected zone; red: visible heat-affected zone).

It can be observed that an even higher coefficient of determination could be identified when correlating the residual stress at the surface with the maximum temperature gradient at the surface rather than the Hollomon–Jaffe parameter or the maximum temperature rate. This corresponds to the expectations, as higher maximum temperature gradients lead to more pronounced plastic flow due to the higher thermally caused stress. As long as the thermally induced plastic flow can be considered the dominant mechanism in terms of the thermal impact of the residual stress, the regression shown in Figure 17 can be applied. As also illustrated in the chart, increasing thermal loads lead to a change of mechanisms that shifts the dominance of the plastic flow due to thermally induced stress toward another mechanism that mainly influences the residual stress at the surface. The metallographic cross-section analyses indicate that this shift is due to annealing effects. In addition, Figure 17 shows the specific grinding power measured during grinding. A correlation between the specific grinding power and the residual stress at the surface cannot be observed. It can be derived that the maximum temperature gradient at the surface (internal material load) is more appropriate to describe the residual stress change at the surface than the specific grinding power (process quantity).

In order to determine the functional relationship between the residual stress change and thermal loads in grinding for processes in which annealing occurs, further examinations will be carried out in the regime of $(\partial T/\partial z)_{max} = 600–1600\ °C/mm$. In this context, the evolution of the microstructure has to be taken into account in order to accurately

describe material modifications during grinding with high thermal loads. This should provide further knowledge about the changing dominance of the mechanisms responsible for the change of the residual stress at the surface.

## 5. Conclusions

In this paper, the hardness reduction and its quantitative dependence on the temperature and temporal development during grinding are shown. Correlations between the hardness change and the maximum temperature rate for different maximum temperatures were determined. As a key result of this paper, the hardness change was described as a function of the incrementally calculated Hollomon–Jaffe parameter, which, in this case, represents the influence of the time-dependent temperature change as well as the respective absolute values of the temperature on the hardness reduction for grinding. The found correlations show a good coefficient of determination between 82% and 99%, depending on whether metallographic cross-sections indicated tempering zones or not. For the first time, a Process Signature component, as a direct connection between material loads and resulting modifications, describing the hardness change during grinding was established.

In addition, the residual stress change at the surface after grinding showed a good coefficient of determination of 84% with the maximum temperature gradient at the surface, leading to another Process Signature component for grinding. This Process Signature component is applicable if the residual stress change is generated due to the plastic flow caused by thermal loads. Increasing thermal loads will result in a change of mechanisms, leading to a higher dominance of tempering effects. Further investigations aim at generating internal material loads in the regime of the expected change of mechanisms in order to extend the Process Signature component into this range of thermal loads generated during grinding.

In order to show the extent and the beginning of changes in residual stress and local hardness for the same internal material loads during grinding, it could be of interest to correlate both modifications. In this context, the correlations should include values of hardness and residual stress changes at the surface since no influence of thermal loads on the residual stress could be observed under most of the grinding parameter combinations. However, Vickers hardness measurement did not allow the hardness to be measured at the surface. Furthermore, different mechanisms led to a change in residual stress and hardness, respectively. Thus, different internal material load quantities have to be considered to explain the development of residual stress and hardness. For these reasons, correlations between residual stress and local hardness are not presented in this paper.

The benefit of Process Signatures in this approach is a better comprehension of processes with thermal impact. Different processes are comparable regarding their thermal impact without explicit consideration of their kinematics. With this approach, the surface integrity of workpieces can be adjusted more easily as the identified correlations will be part of the inversion of the cause-effect relationships from Figure 1, which are required to select appropriate system and process parameters in grinding. The development of this procedure is part of future work, and the Process Signature components identified here are an essential part of this procedure.

**Author Contributions:** Conceptualization, E.K., C.H. and M.E.; validation, investigation, and writing—original draft preparation, E.K.; writing—review and editing, C.H. and M.E.; visualization, E.K.; project administration, C.H. All authors have read and agreed to the published version of the manuscript.

**Funding:** The scientific work has been funded by the Deutsche Forschungsgemeinschaft (DFG, German Research Foundation)—Project-ID 223500200—TRR 136, "Process Signatures", subproject F06.

**Data Availability Statement:** The data presented in this study are available on request from the corresponding author.

**Acknowledgments:** The authors express their sincere thanks to the DFG for funding this project.

**Conflicts of Interest:** The authors declare no conflict of interest.

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
