# Peer review of "Evaluation of Hardness and Residual Stress Changes of AISI 4140 Steel Due to Thermal Load during Surface Grinding"

_jmmp, doi:10.3390/jmmp5030073_

Round 1

Reviewer 1 Report

The reviewer comments of the paper «Evaluation of hardness and residual stress changes due to thermal load during grinding»- Reviewer

The authors presented an article «Evaluation of hardness and residual stress changes due to thermal load during grinding». However, there are several points in the article that require further explanation.

Comment 1:

In the title and abstract, add the research material - AISI 4140. What kind of grinding is investigated in the article?

Comment 2:

The introduction needs to be improved.

First, group citations should be avoided [3-5], [9-11], [12-17], [19-21]. Divide this sentence into several sentences. Clarify the specifics of each article.

Second, it is helpful to add a short paragraph analyzing numerical methods for temperature and grinding processes. It is helpful to add an articles here:

International Journal of Advanced Manufacturing Technology 2017, 91 (9-12), 4055-4068, doi: 10.1007/s00170-017-0036-4

Mechanics of Composite Materials 2015, 51(1), 77-92, doi:10.1007/s11029-015-9478-7

Third, at the end of the introduction, write clearly the "white" spots. That is, that other scientists have not previously investigated this topic.

Comment 3:

  1. Materials and Methods

Add a table of the chemical composition of AISI 4140.

For devices, software and machines used in research, indicate in parentheses (manufacturer, city, country).

Are all the figures original? If not, please give appropriate citations and get permission.

Draw a finite element model of the grinding process in the figure. Is the finite element mesh size uniform or changing? Discuss this in the article.

Describe in more detail how the grinding wheel is modeled? How is the workpiece modeled? What are the boundary conditions?

Give in the table the physicomechanical and thermophysical properties of the material of the workpiece and the grinding wheel. How are they taken into account in the FEM?

What are the characteristics of the PC on which the calculations were carried out? How long did the drilling process take on this PC?

Comment 4:

  1. Experimental and numerical findings

All physical quantities found in formulas must be named. Poets carefully check the correspondence to this.

Figures 5, 7, 13 show the fractions "," instead of "." Fix it.

Comment 5:

It will be useful to add a section of Nomenclature in which to sign all the physical quantities and abbreviations encountered in the article. There are many physical quantities in the text and such a section will help to find the description of the necessary element.

For example,

lw               : Workpiece length (mm)

FEM         : Finite element method  

etc.

Comment 6:

Conclusions.

Draw quantitative and qualitative conclusions.

In addition, it is necessary to more clearly show the novelty of the article and the advantages of the proposed method. What is the difference from previous work in this area? Show practical relevance. Conclusions should reflect the purpose of the article.

Comment 7:

The article must be carefully proofread by a native English speaker.

The article is interesting. Authors should carefully study the comments and make improvements to the article. After major changes can an article be considered for publication in the «Journal of Manufacturing and Materials Processing».

Reviewer 2 Report

The manuscript entitled: 'Evaluation of hardness and residual stress changes due to thermal load during grinding' deals with the variation in hardness in residual stresses due to the thermal load induced during the grinding process. My concerns are:

(1) How was the residual stress profile calculated?

(2) XRD and/or synchrotron should be used to measure the residual stress states

(3) There is no scientific discussion correlation the residual stress w.r.t local harndess.

(4) Overall the manuscript lacks scientific depth/discussion

Reviewer 3 Report

The paper deals with an important practical engineering problem in the field of manufacturing and materials processing.

It is a detailed piece of work of enough scientific quality to be published.

Reviewer 4 Report

This paper aims to develop a functional relationship between material modifications and thermal material loads during grinding. This can be used to make a prediction before choosing a grinding process. However, there are some question need to be explained :

  1. The change of hardness or residual stress is usually caused by the evolution of microstructure. In order to accurately describe the relationship between material modification and thermal loading, it will be better to consider the evolution process and the amount of phase transformation, but the relevant content is not given in this paper.
  2. Using simulation to obtain the peak temperature, temperature gradient and heating rate is a feasible method. For this method, the simulation accuracy is very important, and relying solely on literature data may not be applicable. Especially, the subsequent analyses are based on the results of temperature simulation. The calibration process should therefore be described in detail.

  3. It has been mentioned in the paper that different grinding processes have a negligible impact on residual stress and hardness. This is mainly resulted by different process parameters leading to different thermal loadings. However, the relationship between grinding processes and thermal loading should be discussed. It may be helpful to explain change of hardness or residual stress. The whole temperature change of different processes can be measured and simulated.

  4. For Figure10 and Figure11, data points seem to be insufficient. Will the number of data points affect the subsequent parameter fitting?

  5. The format and language of this paper should be improved.

Round 2

Reviewer 1 Report

The authors have improved the article according to the comments. However, before accepting it is important for the authors to add in tabular form "The properties of the workpiece were used according to Richter [28]." It is not enough just citations. Therefore, I return you to an early comment:
Give in the table the physicomechanical and thermophysical properties of the material of the workpiece and the grinding wheel.
Authors must fully disclose to the reader all the nuances of the posed problem.

Reviewer 2 Report

I do not see any significant changes in the manuscript in terms of scientific quality and hence cannot recommend the manuscript for publication. In addition, the comments raised were not answered to the point.

Reviewer 4 Report

It is a detailed  work and could be considered to publish.

Round 3

Reviewer 1 Report

The article can now be accepted for publication.

Reviewer 2 Report

The authors have somehow managed to satisfy my queries and so I recommend the manuscript for publication in the present form.